# Sustainability of Shipping Logistics: A Warning Model

**Ronghua Xu** [1] , **Yiran Liu** [2], **Meng Liu** [3,*] and **Chengang Ye** [2]

1  Business School, Ningbo University, Ningbo 315211, China
2  Business School, University of International Business and Economics, Beijing 100029, China
3  School of International Business and Management, Sichuan International Studies University, Chongqing 400031, China
*  Correspondence: mengliu@sisu.edu.cn

**Abstract:** The shipping industry is the foundation of the economy, and it is affected by fluctuations in the economic cycle. The mainstream of financial early warning research is quantitative modeling research. There are few systematic studies on financial early warning of shipping enterprises, and most of them still remain in the qualitative stage. This paper chooses Chinese listed shipping companies as its target, takes the economic cycle as an important reference, and then uses logistic regression, neural network, and random-forest methods to establish a model for financial warning. The random-forest model is employed to rank the importance of warning indicators. The results show that it is effective to consider macro-factors, such as the economic cycle, and the predictive accuracy of the random-forest method is higher than that of the financial warning models established by logistic regression and by the neural network. Financial alerts can help managers prepare for crises in advance. The purpose of this paper is to provide an early warning model for the sustainable development of shipping logistics.

**Keywords:** shipping enterprises; economic cycle; financial early warning; random forest

## 1. Introduction

The shipping industry is the foundation of the economy. From an economic and military point of view, ships in times of peace can be used in the sailing trade, and ships in times of war can be converted into warships. The historical high point of the BDI (Baltic Dry Index), on 20 May 2008, was 11,793 points. After the outbreak of the economic crisis, the shipping industry suffered a downturn, and the BDI hit a historical low of 290 on 10 February 2016, a difference of 40 times in about eight years. Affected by the epidemic in 2020, the shipping industry has recovered once again during the last two years, and the BDI had risen to more than 5000 points in October 2021 [1].

This shows that the shipping industry is an industry that is relatively influenced by the external economy and that early warning of financial risk to shipping enterprises is very important. Shipping enterprises need a risk early warning management system that can take macro-factors into account so that managers can foresee potential financial risks that may occur in the future in advance in order to be fully prepared to deal with oncoming risks [2]. As China is the world's largest cargo trading country, its shipping industry is representative, so this paper selects Chinese listed shipping companies as its research object.

Mainstream financial early warning research is quantitative model research. Systematic research on the financial early warning of shipping enterprises has been less studied, and most of it still remains in the qualitative stage. Compared with qualitative analysis, the quantitative model quantifies the relevant factors that affect industry enterprises, which are more easily operated and used by the shipping enterprises. Therefore, this paper selects the method of establishing quantitative mathematical models to provide an early warning for the financial situations of shipping enterprises. The three early warning models used in this paper are as follows: logistic regression, the neural network, and random forest.

Shipping enterprises are influenced by economic cycles, so it is necessary to add the factor of economic cycles when constructing a financial early warning model. Meanwhile, the shipping industry itself has the characteristics of seasonality, profitability, service, and capital intensity, so it is also necessary to fully consider both the macro- and micro-factors that affect the operation of shipping enterprises when constructing the financial early warning model. The economic cycle factor is mainly reflected in the selection of the sample time period. In this paper, we establish a model based on samples from 2002–2009, and then we use the samples from 2010–2015 to test the established model. According to the division of economic cycles used in this paper, the modeling sample data from 2002–2009 cover a whole economic cycle in terms of time period; this makes the model more objective and comprehensive [3].

Among the three models, the random-forest financial early warning model can best analyze the importance of the input raw variables and has the best forecasting ability and operability. With the inclusion of macro-variables, the random-forest model achieves 100% and 80.5% accuracy for the training and testing samples, respectively.

This article contributes to the existing literature as follows. From the research content, the macro-economic factors that affect shipping enterprises are considered; from the research perspective, the factors that affect economic cycles on enterprise finances are added to the model; and from the research method, the random-forest model is used for the importance analysis of variables. Thus, a financial early warning model that integrates economic cycles, considers macro-factors and micro-factors, and combines the characteristic indicators of shipping enterprises is established.

The structure of the paper is as follows. Section 2 introduces the three financial early warning models, including the logistic regression, neural network, and random-forest models. Section 3 illustrates sample selecting procedure. Section 4 interprets the process of selecting indicators. Section 5 trains and tests the proposed warning models. Section 6 addresses the limitations and suggestions. Section 7 concludes the paper.

## 2. Financial Early Warning Model

### 2.1. Literature Review

Financial forecasting is a financial management method that warns of possible future business risks through the capture and analysis of corporate financial indicators. This enables enterprises to take early measures to avoid crises and reduce losses before they occur or to plan in advance how to respond when a crisis occurs. It is of great significance to the development of enterprises.

There are two main types of financial early warning research methods: qualitative methods and quantitative methods. In the early years, scholars preferred to use qualitative methods for financial early warning research. These methods mainly analyze the "internal and external problems" of enterprises from a qualitative perspective to help enterprises identify risk points and avoid crises. In recent years, quantitative analysis has gradually become the mainstream of financial early warning research, and various research methods and model improvements have emerged. Models with high prediction accuracy are better.

This paper argues that the current financial early warning model still has the following problems.

First, most scholars in the article distinguish between financial crises as ST (*ST) and non-ST (non-*ST), i.e., two consecutive years of losses; however, in reality, there are often listed companies that engage in surplus manipulation, and in order not to be affected by ST in terms of stock price and market confidence, the first financial year loss is followed by a series of non-operating activities in the second financial year to adjust profits. Therefore, this paper argues that attention and caution should be drawn when a company has a loss.

The second problem is the sample time period selection. In some domestic scholars' research, the samples for establishing a financial early warning model and testing said financial early warning model often use the data on the same time period, and the data on

the same time period have certain similarities and approximations, thus a higher prediction accuracy rate can be obtained.

Thirdly, research on financial early warning model for shipping enterprises is still relatively minimal, and only internal financial factors are often considered in the model research. In the models constructed by Wang (2005) [2] and Yang (2010) [4], there are more factors that require subjective judgment; thus, they are not objective enough. Although the prediction accuracy of the model obtained by Ai-Ping Gan (2014) [5] reached 100%, its data volume was small, covered a short period of years, did not take into account macro-economic factors, and did not set test variables to test the model.

Fourth, the financial early warning model does not take into account the economic cycle factor. No one has taken into account the economic cycle factor in the current financial early warning model. For the shipping industry, which is influenced by fluctuations in the economic cycle, the economic cycle should be taken into account when constructing the financial early warning model.

To address these problems in the current literature, this paper innovates indicator screening and indicator interval selection.

The models used in this paper are shown in Sections 2.2–2.4.

### 2.2. Intoroduction of Logistic Model

The logistic regression model is a linear regression model obtained by performing logit transformation on the probability $\pi(Y = 1)$. Compared with the regular regression model, the left side of a logistic regression model equation is not the continuous dependent variable Y, but the logit transformed value of the probability $\pi(Y = 1)$ for $Y = 1$.

$$\log it[\pi(Y = 1)] = \ln[\frac{\pi(Y = 1)}{1 - \pi(Y = 1)}] = \beta_0 + \beta_1 X_1 + \beta_2 X_2 + \ldots + \beta_m X_m + \varepsilon \quad (1)$$

The independent variables are $X_1$, $X_2$, $\ldots$ $X_m$.$\exp(*)$ is the exponential with the base of the natural logarithm. $\beta_0$ is the intercept (constant term). $\beta_j$ is $X_j (j = 1, 2, \ldots, m)$, the partial regression coefficient. The corresponding logistic regression model is:

$$\pi(Y = 1) = \frac{\exp(\beta_0 + \beta_1 X_1 + \beta_2 X_2 + \ldots + \beta_m X_m)}{1 + \exp((\beta_0 + \beta_1 X_1 + \beta_2 X_2 + \ldots + \beta_m X_m))} \quad (2)$$

The financial crisis probability of a company can be calculated as follows: First, obtain the value of the two variables: the financial early warning variable $X_j (j = 1, 2, \ldots, m)$ and partial regression coefficient $(b_0, b_1, \ldots, b_j)$. Then, plug it into the Formula (3) and estimate $\hat{P}$. $\hat{P}$ is the probability of a financial crisis of the company [6].

$$\hat{P} = \frac{\exp(b_0 + b_1 x_1 + b_2 x_2 + \ldots + b_m x_m)}{1 + \exp(b_0 + b_1 x_1 + b_2 x_2 + \ldots + b_m x_m)} \quad (3)$$

### 2.3. Intoroduction of Neural Network Model

A multilayer neural network has three layers: an input layer, a hidden layer, and an output layer. For the financial early warning model in this paper, the input layer is the extracted early warning variables, and the input sample dimension is the number of early warning variables, which is four in the early warning model that only considers microscopic factors and four after adding macro-variables [7]. The hidden layer is mainly some nonlinear transformation functions f($*$). Generally speaking, the more complex the hidden layer, the higher the accuracy of the training samples. The output layer is the output of the neural network model. The output layer usually uses a linear transformation function; that is, it uses an activation function. The commonly used activation functions are:

Logistic function, $f(z) = \frac{1}{1 + e^{-z}}$, the value range (0, 1).

The hyperbolic tangent function, $f(z) = \frac{e^z - e^{-z}}{e^z + e^{-z}}$, takes the value range $(-1, 1)$.

### 2.4. Intoroduction of Random-Forest Model

Random forest is obtained by integrating multiple decision trees; it is a machine-learning algorithm based on classification trees. The basic idea and principle of random forest is that when the variable is input, each tree in the random forest will discriminate and classify the sample; finally, it will collect the classification results of each tree, and each tree will vote to select the class with the most results. That is the model classification of the sample [8].

The financial early warning model in this paper uses the algorithms of CART and bagging for classification. The specific algorithm process of bagging is as follows:

First, perform bootstrap sampling on the training sample set to obtain a new training sample set $L_m(m = 1, 2, \dots M)$ with a sample size of N and construct a decision tree.

Secondly, combine M decision trees $h_B(x; L_m)$, then obtain the final classifier $H_B$. The prediction of $H_B$ is $argmax_j N_j$. Among them, $N_j = \sum_{m=1}^{M}\{I(h_B(x; L_m) = j)\}$, $I(*)$ is the indicative function.

The random-forest classification model is composed of a large number of decision trees and bagging classification models $\{h(X, \theta_k), k = 1, 2, \dots, N\}$. $\theta_k$ is an independent and identically distributed random vector. After k several cycles, the final classification decision model of the random forest is as follows [9]:

$$H(x) = \operatorname*{argmax}_{Y} \sum_{i} I(h_i(x) = Y) \tag{4}$$

Finally, the permutation importance of the variable can be obtained as follows:

$$\bar{d} = \frac{1}{|T|} \sum_{t \in T} R_t - R_t^* \tag{5}$$

## 3. Sample Selection

### 3.1. Definition of Economic Cycle

The shipping market is a trade-derived market, and the level of trade development depends on economic demand, so the financial status of shipping companies is closely linked to the economic cycle. When the economy is up, trade demand increases, orders increase, and shipping companies are in a good financial condition; when the economy is down, trade demand falls, orders decrease, and shipping companies are in a poor financial condition or even lose money [10].

British economist Stopford (1988) [1] pointed out in his published book *Maritime Economics* that the shipping market also has its own cycle, which can be divided into the seasonal cycle, the short cycle, and the long cycle by analogy to the economic cycle. This paper mainly considers an economic cycle of 5–10 years, which is more consistent with the shipping market cycle. It has a slow change speed and is easier to grasp in order to analyze the shipping enterprises. Scholars often use the GDP (gross domestic product) growth rate to measure the economic cycle [11]. Combined with the BDI index, GDP trends, and previous scholars' definition of China's economic cycle, this paper's definition of the economic cycle is shown in Figure 1.

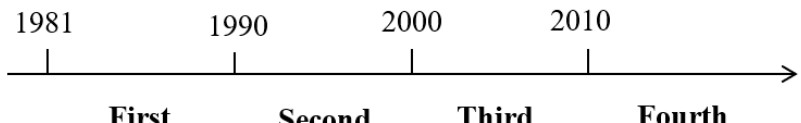

**Figure 1.** Division of China's economic cycle since the reform and opening up.

### 3.2. Sample Interval

The empirical ideas of this paper are to build a model based on the data of a whole economic cycle and to use the data outside this economic cycle as a test variable to verify

the accuracy and validity of the model. This paper selects shipping companies' statistics from 2002 to 2009 as the modeling sample interval and statistics from 2010 to 2015 as the inspection sample interval [12].

### 3.3. Sample Definition

This paper selects Chinese listed shipping companies with negative net profits as crisis samples, then defines 1 as a crisis sample and 0 as a healthy company sample.

It sets year t as the year when the company was ST (special treatment). Previous scholars often select the data of the t-3 year for financial early warning research when conducting research. This is because the listed company had two years of losses before ST (*ST), namely t-1 and t-2. There are very few shipping companies that are ST (*ST), so this paper defines the year when the net profit of shipping companies is negative as year t and uses the macro- and micro-data of year t-1 to make predictions to improve the accuracy of the model [13]. And it is assumed that this is the company's first time suffering a loss; there has been no sign of loss before.

### 3.4. Sample Matching Method

The main matching factors considered in this paper are fiscal year and sample size. The healthy sample companies need to be financially healthy in the year when the crisis sample companies lose money.

In previous financial early warning research, the proportion of crisis companies and healthy companies in the sample is about 1:1–1:3. The number of crisis companies among Chinese listed shipping companies is small. In order to ensure a sufficient number of samples, in this paper, a random unpaired sample of crisis companies and healthy companies is selected in an annual matching of roughly 1:3 [14].

### 3.5. Sample Establishment

From the currently listed shipping companies in China, excluding the companies with incomplete data and years, the training sample is selected, including 80 shipping companies with different fiscal years from 2002–2009, and it includes 22 crisis company samples and 58 healthy company samples. The test sample is selected from 77 shipping companies in different fiscal years from 2010–2015, including 21 crisis company samples and 56 healthy company samples. The data in this paper are obtained from the Wind Economic Database.

## 4. Indicator Selection

### 4.1. Primary Selection of Micro-Indicators

Combined with the indicators selected by the research conducted by experts and scholars in recent years, this paper initially selects 20 indicators from five aspects: profitability, short-term solvency, long-term solvency, operating ability, and growth ability. The specific 20 micro-financial early warning indicators are shown in Table 1.

The 20 indicators basically and comprehensively reflect the financial status of the enterprises, and this paper considers them to be representative. In order to ensure the sensitivity of the indicators, the indicators should be able to reflect the differences between crisis enterprises and healthy enterprises, and further screening of indicators is required [15].

First of all, it is necessary to judge whether there is a difference between the two groups of indicators. If there is no difference in an indicator, that is, if the result is not significant, it should be eliminated. The commonly used significance test methods are the t test and the Mann–Whitney test. The t test requires the data to be tested should exhibit a normal distribution; the Mann–Whitney test can test the significance of non-normal data. Therefore, before the significance test, this paper selects the Kolmogorov–Smirnov test to test the normal distribution of the data. The K–S test is bounded by 0.05. When the test result is greater than 0.05, the variable obeys the normal distribution; otherwise, it does not obey the normal distribution [16].

**Table 1.** Primary selection of financial indicators.

|  |  | Indicator Name | Calculation Formula |
|---|---|---|---|
| Profitability | X1 | Basic earnings per share | Net profit/total number of shares |
| | X2 | Net interest rate on total assets | Net profit/average net assets |
| | X3 | ROA | (Total profit + interest expense)/total average assets |
| | X4 | Operating income/operating profit | |
| | X5 | Sales margin | Net profit/operating income |
| Short-term solvency | X6 | Current ratio | Current assets/current liabilities |
| | X7 | Quick ratio | (Current assets—inventory)/current liabilities |
| | X8 | Cash flow interest coverage ratio | Cash flow/interest expense from operating activities |
| | X9 | Earned Interest Multiple | EBIT/Interest expense |
| Long-term solvency | X10 | Long-term debt ratio | Total long-term liabilities/assets |
| | X11 | Assets and liabilities | Total liabilities/total assets |
| | X12 | Equity ratio | Total liabilities/shareholders' equity |
| Operating capacity | X13 | Inventory turnover | Operating cost/annual average inventory |
| | X14 | Accounts receivable turnover | Operating income/annual average revenue accounts |
| | X15 | Accounts payable turnover | Main business cost/annual average accounts payable |
| | X16 | Total asset turnover | Operating income/annual average assets |
| Growth ability | X17 | Basic earnings per share (year-over-year growth rate) | Current year–last year/last year |
| | X18 | Total operating income (year-on-year growth rate) | |
| | X19 | Operating profit (year-on-year growth rate) | |
| | X20 | Net profit (year-on-year growth rate) | |

The Mann–Whitney test and the t test make assumptions about the overall sample variables in advance, then calculate the statistical $p$ values according to the sample, and finally compare the results with the null hypothesis. The specific steps are as follows:

①  Make a hypothesis $H_0$: Two independent samples come from the same population; $H_1$: Two independent samples come from different populations.

②  Calculated $p$ value.

③  If $p > \alpha$, accept $H_0$, otherwise $H_0$ reject $H_1$.

For the indicators that pass the significance test, in order to remove duplicate information, improve the computational efficiency of the early warning model, and facilitate modeling, the commonly used principal component method is used to reduce the dimension [17]. This is the overall micro-index screening idea, as shown in Figure 2.

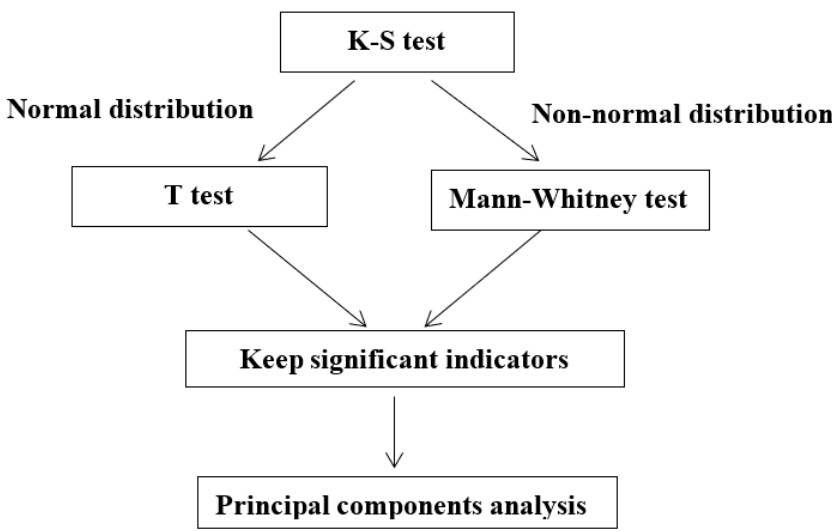

**Figure 2.** The way to screen micro-indicators.

### 4.2. Kolmogorov–Smirnov Normal Distribution Test

This paper uses SPSS to carry out the K–S normal distribution test. After inputting two groups of sample data, the results are shown in Table 2.

**Table 2.** Kolmogorov–Smirnov test.

| Group | 0 | | 1 | |
|---|---|---|---|---|
| | Test Statistics | Progressive Significance | Test Statistics | Progressive Significance |
| X1 | 0.179 | 0.000 | 0.305 | 0.000 |
| X2 | 0.161 | 0.001 | 0.362 | 0.000 |
| X3 | 0.167 | 0.000 | 0.361 | 0.000 |
| X4 | 0.094 | 0.200 * | 0.514 | 0.000 |
| X5 | 0.098 | 0.200 * | 0.519 | 0.000 |
| X6 | 0.180 | 0.000 | 0.141 | 0.200 * |
| X7 | 0.168 | 0.000 | 0.127 | 0.200 * |
| X8 | 0.122 | 0.031 | 0.223 | 0.006 |
| X9 | 0.325 | 0.000 | 0.403 | 0.000 |
| X10 | 0.090 | 0.200 * | 0.201 | 0.021 |
| X11 | 0.138 | 0.007 | 0.249 | 0.001 |
| X12 | 0.191 | 0.000 | 0.485 | 0.000 |
| X13 | 0.205 | 0.000 | 0.362 | 0.000 |
| X14 | 0.199 | 0.000 | 0.197 | 0.026 |
| X15 | 0.382 | 0.000 | 0.301 | 0.000 |
| X16 | 0.232 | 0.000 | 0.142 | 0.200 * |
| X17 | 0.452 | 0.000 | 0.267 | 0.000 |
| X18 | 0.119 | 0.040 | 0.216 | 0.009 |
| X19 | 0.197 | 0.000 | 0.366 | 0.000 |
| X20 | 0.403 | 0.000 | 0.261 | 0.000 |

* indicates significance level greater than 0.05.

In group 0, the significance level of $X_4$, $X_5$, and $X_{10}$ exceeds 0.05. In group 1, the significance level of $X_6$, $X_7$, and $X_{16}$ exceeds 0.05. In group 0 and 1, no indicator significance level exceeds 0.05 at the same time; this means that none of the 20 financial variables conform to a normal distribution [18].

*4.3. Significance Test*

Since none of the variables obey the normal distribution, the t test method is discarded in this paper, and the Mann–Whitney test method is used to analyze the difference between the two groups. The data were input into SPSS, and the results are shown in Table 3.

**Table 3.** Mann–Whitney U test.

| Variable | Mann–Whitney U Statistical Data | Wilcoxon W | Z | Salience |
|---|---|---|---|---|
| X1 | 111 | 364 | −5.679 | 0.000 |
| X2 | 100 | 353 | −5.798 | 0.000 |
| X3 | 113 | 366 | −5.657 | 0.000 |
| X4 | 53 | 306 | −6.304 | 0.000 |
| X5 | 57 | 310 | −6.261 | 0.000 |
| X6 | 598 | 2309 | −0.431 | 0.666 * |
| X7 | 593.5 | 2304.5 | −0.480 | 0.632 * |
| X8 | 208 | 461 | −4.634 | 0.000 |
| X9 | 82 | 335 | −5.992 | 0.000 |
| X10 | 399 | 652 | −2.575 | 0.010 |
| X11 | 227 | 1938 | −4.429 | 0.000 |
| X12 | 556 | 2267 | −0.884 | 0.377 * |
| X13 | 351 | 2062 | −3.093 | 0.002 |
| X14 | 322 | 575 | −3.405 | 0.001 |
| X15 | 601 | 854 | −0.399 | 0.690 * |
| X16 | 615 | 2326 | −0.248 | 0.804 * |
| X17 | 363 | 616 | −2.963 | 0.003 |
| X18 | 248 | 501 | −4.203 | 0.000 |
| X 19 | 404 | 657 | −2.522 | 0.012 |
| X 20 | 384 | 637 | −2.737 | 0.006 |

* indicates significance level greater than 0.05.

It can be seen from Table 3 that the four index variables $X_6$ (current ratio), $X_7$ (quick ratio), $X_{15}$ (accounts payable turnover ratio), and $X_{16}$ (total asset turnover ratio) are greater than 0.05, and they fail the significance test. There is no significant difference between the crisis and healthy enterprise samples among these four variables, and they are not sensitive, so they are excluded.

*4.4. Principal Component Analysis*

Principal component analysis of variables can eliminate the correlation between variables, simplify the variables in the model, and reduce the complexity of the model. After the above screening, there are 16 financial indicators entering the step of principal component analysis [19]. Before principal component analysis, the KMO and Bartlett tests should be performed on the factors to determine whether the variables can extract the principal components. Generally speaking, when the KMO value is close to 1, it means that the principal components can be extracted; when the significance level is lower than 0.05, it means that the indicators are correlated with each other and that the variables are more suitable for principal component analysis. The 16 financial indicators of the 80 modeling samples were imported into SPSS, and the test results obtained are shown in Table 4.

**Table 4.** KMO and Bartlett Test.

| KMO and Bartlett Test | | |
|---|---|---|
| Kaiser–Meyer–Olkin Measurement of sampling suitability | | 0.692 |
| | Approximately chi-square | 1803.374 |
| Bartlett's Sphere Test | Degree of freedom | 105 |
| | Salience | 0.000 |

As can be seen from Table 4, the KMO value is 0.692, indicating that the principal components can be extracted from 16 financial variables, and the principal component analysis of these 16 variables is effective. The significance of Bartlett's sphericity test is 0.000, which is less than 0.05, indicating that the remaining 16 variables are correlated with each other. Through principal component analysis, the correlation between financial variables can be eliminated.

Next, this paper identifies and interprets the principal components of the 16 financial indicators [20]. Referring to the practice of previous scholars, this paper extracts the first four principal components with eigenvalues greater than 1, as shown in Table 5, and their cumulative contribution rate reaches 72.5%.

**Table 5.** Coefficient of variation statistics.

| Element | Starting Eigenvalues | | | Extract Sum of Squares and Load | | |
|---|---|---|---|---|---|---|
| | Total | Mutations % | Cumulative % | Total | Mutations % | Cumulative % |
| 1 | 5.915 | 39.430 | 39.430 | 5.915 | 39.430 | 39.430 |
| 2 | 2.212 | 14.745 | 54.175 | 2.212 | 14.745 | 54.175 |
| 3 | 1.694 | 11.296 | 65.471 | 1.694 | 11.296 | 65.471 |
| 4 | 1.057 | 7.045 | 72.516 | 1.057 | 7.045 | 72.516 |
| 5 | 0.931 | 6.207 | 78.723 | | | |
| 6 | 0.861 | 5.741 | 84.465 | | | |
| 7 | 0.693 | 4.620 | 89.085 | | | |
| 8 | 0.669 | 4.461 | 93.545 | | | |
| 9 | 0.382 | 2.547 | 96.093 | | | |
| 10 | 0.299 | 1.991 | 98.083 | | | |
| 11 | 0.138 | 0.918 | 99.002 | | | |
| 12 | 0.116 | 0.774 | 99.776 | | | |
| 13 | 0.032 | 0.213 | 99.989 | | | |
| 14 | 0.002 | 0.011 | 100.000 | | | |
| 15 | 0.000 | 0.000 | 100.000 | | | |

The first principal component has the largest load on the variables $X_1$, $X_2$, $X_3$, $X_4$, and $X_5$, and these five indicators all represent the profitability of shipping companies. The expression for the first principal component is as follows:

$$
\begin{aligned}
FAC_1 = 0.154X_1 \quad &+ 0.163X_2 + 0.164X_3 + 0.136X_4 + 0.136X_5 + 0.085X_8 + 0.055X_9 \\
&+ 0.054X_{10} - 0.134X_{11} - 0.041X_{13} + 0.063X_{14} + 0.034X_{17} \quad (6) \\
&+ 0.094X_{18} + 0.076X_{19} + 0.046X_{20}
\end{aligned}
$$

The second principal component has a relatively large load on the $X_{17}$, $X_{19}$, and $X_{20}$ variables; these three variables all reflect the growth ability of the enterprise. The expression for the second principal component is as follows:

$$
\begin{aligned}
FAC_2 = {} &-0.05X_1 - 0.037X_2 - 0.042X_3 - 0.155X_4 - 0.158X_5 + 0.036X_8 \\
&- 0.032X_9 + 0.064X_{10} + 0.089X_{11} - 0.17X_{13} + 0.111X_{14} \quad (7) \\
&+ 0.359X_{17} + 0.067X_{18} + 0.274X_{19} + 0.366X_{20}
\end{aligned}
$$

The third principal component on $X_8$ and $X_{13}$ is relatively large, which mainly reflects the short-term solvency of the enterprise. The expression for the third principal component is as follows:

$$
\begin{aligned}
FAC_3 = {} &-0.039X_1 + 0.031X_2 + 0.033X_3 + 0.222X_4 + 0.224X_5 - 0.375X_8 \\
&- 0.148X_9 - 0.211X_{10} - 0.016X_{11} + 0.252X_{13} - 0.311X_{14} \quad (8) \\
&+ 0.243X_{17} - 0.058X_{18} + 0.002X_{19} + 0.238X_{20}
\end{aligned}
$$

The fourth principal component has a large load on $X_{10}$ and $X_{14}$, which is the embodiment of the operational capability of the shipping enterprise. The expression of the fourth principal component is as follows:

$$\begin{aligned}
FAC_4 = 0.097X_1 \quad & + 0.037X_2 + 0.067X_3 - 0.056X_4 - 0.056X_5 + 0.378X_8 \\
& - 0.149X_9 - 0.589X_{10} + 0.129X_{11} + 0.452X_{13} + 0.415X_{14} \\
& + 0.113X_{17} + 0.03X_{18} - 0.099X_{19} + 0.042X_{20}
\end{aligned} \quad (9)$$

*4.5. Primary Selection of Macro-Indicators*

Based on the indicators referenced by the China shipping prosperity index and the ease of obtaining data, this paper selects 11 macro-warning indicators, as shown in Table 6.

**Table 6.** Primary selection of macro-variables.

| Variable | Variable Name | Data Sources |
|---|---|---|
| M1 | BDI | Wind information |
| M2 | CCFI: Composite Index | Wind information |
| M3 | The total amount of waterborne freight transport nationwide (100 million tons) | Zhonghong Industry Database |
| M4 | Annual cargo throughput of major coastal ports (100 million tons) | Zhonghong Industry Database |
| M5 | Demand deposit rate | Wind information |
| M6 | Annual GDP growth rate | National Bureau of Statistics of China |
| M7 | RMB to USD exchange rate (USD = 100) (yuan) | National Bureau of Statistics of China |
| M8 | National power generation (100 million kw) | Zhonghong Industry Database |
| M9 | CPI | wind information |
| M10 | RPI | National Bureau of Statistics of China |
| M11 | National annual import and export value (100 million yuan) | National Bureau of Statistics of China |

*4.6. Principal Component Analysis*

Referring to the screening of the micro early warning indicators described above, this section also uses SPSS to conduct principal component analysis and extraction of the macro early warning indicators. First, it is necessary to use KMO and Bartlett to analyze whether these 11 macro early warning indicators are suitable for principal component analysis. The KMO value is 0.618, indicating that the principal components can be extracted from the 11 variables; the significance level of the Bartlett sphere test is 0.000, which is less than 0.05, indicating that there is a certain correlation between the variables. The dimensional reduction analysis can be carried out through the principal components [21].

The principal components are selected from the three principal components whose eigenvalues are greater than 1 in the coefficient of variation statistics table. The cumulative contribution rate of these three principal components is 95.32%, indicating that these 11 macro-variables can be well represented.

The first principal component has relatively large loads on the $M_3$, $M_4$, $M_5$, $M_6$, $M_7$, $M_8$, and $M_{11}$ variables, and these indicators mainly reflect the domestic macro-economic situation.

$$\begin{aligned}
FAG_1 = -0.053M_1 & - 0.003M_2 + 0.137M_3 + 0.139M_4 - 0.126M_5 + 0.139M_6 \\
& - 0.138M_7 + 0.139M_8 + 0.056M_9 + 0.06M_{10} + 0.139M_{11}
\end{aligned} \quad (10)$$

The second principal component has a relatively large load on $M_9$. $M_9$ and $M_{10}$ and reflects the domestic macro-price level.

$$\begin{aligned}
FAG_2 = 0.323M_1 & + 0.219M_2 - 0.023M_3 - 0.005M_4 + 0.161M_5 - 0.031M_6 \\
& + 0.04M_7 + 0.011M_8 + 0.409M_9 + 0.385M_{10} + 0.031M_{11}
\end{aligned} \quad (11)$$

The third principal component on $M_1$ and on $M_2$ is relatively large, which mainly reflects the general environment of domestic shipping enterprises.

$$FAG_3 = 0.368M_1 + 0.609M_2 + 0.122M_3 + 0.089M_4 - 0.017M_5 + 0.079M_6$$
$$+ 0.016M_7 + 0.084M_8 - 0.29M_9 - 0.333M_{10} + 0.045M_{11} \tag{12}$$

## 5. Training and Testing of Warning Model

### 5.1. Training and Testing of Logistic Model

Considering only the micro-financial indicators and inputting the obtained values of the four principal component variables into SPSS to construct a logistic regression model, it can be concluded that the logistic early warning regression model of China's listed shipping companies only considers the micro-financial factors:

$$P_i = \frac{\exp(-1.128 - 7.691FAC_1 - 0.727FAC_2 - 0.656FAC_3 + 0.346FAC_4)}{1 + \exp(-1.128 - 7.691FAC_1 - 0.727FAC_2 - 0.656FAC_3 + 0.346FAC_4)} \tag{13}$$

Since the sample size ratio used in this paper is about 1:3, a split value of 0.25 is selected as the dividing point between the crisis sample and the healthy sample.

The logistic regression early warning model with macro-factors added is:

$$P_i = \frac{\exp(-3.047 - 8.574FAC_1 - 1.022FAC_2 - 0.562FAC_3 + 0.217FAC_4 - 2.503FAG_1 + 2.175FAG_2 - 0.534FAG_3)}{1 + \exp(-3.047 - 8.574FAC_1 - 1.022FAC_2 - 0.562FAC_3 + 0.217FAC_4 - 2.503FAG_1 + 2.175FAG_2 - 0.534FAG_3)} \tag{14}$$

### 5.2. Training and Testing of Neural Network Model

This paper uses MATLAB programming to create a three-layer neural network. The input layer comprises four principal components of micro-factors and three principal components of macro-factors. After many tests, the hidden layer is determined to be 10. The number of output layer nodes is 1. The main idea is as follows: first import the training samples, train the network, and then substitute the test samples into the network for simulation [22].

### 5.3. Training and Testing of Random-Forest Model

Since the random forest itself can score the importance of the variables without dimensionality reduction and the maximum number of independent variables that can be input by the random forest is 53, this paper uses the original 20 micro-financial variables that have not undergone principal component analysis and the 11 macro-variables are input into MATLAB, and the number of training samples is still 80. The maximum number of trees that can be constructed in a random forest is 500. Let the number of trees be ntree, and ntree converges at 50 [23]. Finally, the classification results, OOB error, and importance level of variables can be obtained. The importance levels are shown in Figure 3.

### 5.4. Comparison of Model Prediction Results

The correct rates of the three modeling methods used in this chapter are shown in Table 7. Overall, whether it is a training sample or a test sample, the correct rate of modeling with macro- and micro-factors is higher than that of modeling only considering micro-factors. The modeling results of the three methods show that the neural network model is better than the logistic model, and the random-forest model is better than the neural network model [24].

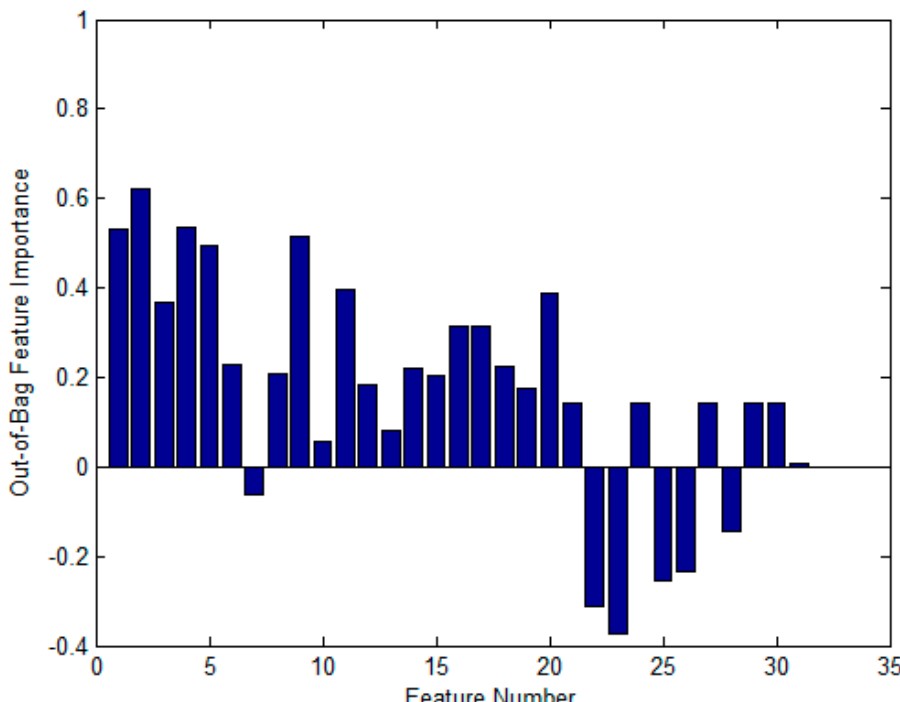

**Figure 3.** Variable importance distribution diagram (micro + macro).

**Table 7.** The comprehensive prediction accuracy of the three models.

| | Training Sample Accuracy | | Testing Sample Accuracy | |
|---|---|---|---|---|
| | **Micro** | **Macro + Micro** | **Micro** | **Macro + Micro** |
| Logistic | 85% | 88.80% | 72.73% | 79.20% |
| Neural networks | 90% | 91.25% | 72.73% | 74.03% |
| Random forest | 100% | 100% | 76.62% | 80.52% |

## 6. Limitations and Discussion

This paper constructs a financial early warning model of Chinese listed shipping enterprises with reference to a large number of previous studies, but the model still has many places where it can be improved for further investigation by subsequent scholars.

In the selection of the sample time period, this paper divides the economic cycle of China since the reform and opening up with what some scholars consider as the medium cycle, and it selects a whole medium cycle of data for prediction. Economic cycles can also be divided into long cycles and short cycles, and future scholars can conduct more detailed research by other divisions of economic cycles [25–27]. The samples used in this paper are all listed shipping enterprises in China; no unlisted shipping enterprises are considered. It is also possible to collect financial information about unlisted enterprises to expand the sample size and to carry out the establishment of a financial early warning model with broader coverage.

In the selection of variables, micro-factors and macro-factors are considered in this paper. However, due to limited time and effort, the selected indicators may not completely cover all of the aspects of the enterprises; other variables can continue to be added to obtain a better model. In the variable time selection, this paper selects the data of year t-1. It can also select the data of more advanced time, and through comparison, the year with the best financial early warning effect for shipping enterprises can be derived [28].

In terms of model selection, the models chosen in this paper are logistic regression, the neural network, and the random-forest model, which have been applied to financial early warnings in recent years. A good financial early warning model should not only have a

good forecasting effect, but also be easy to operate. The model should be simple and clear. In the comparison of the three models, logistic regression and random-forest models are easier to operate. More concise early warning models may be available in the future.

Of course, there are some other models applied to financial early warning, such as the support vector machine model, the CART model, the genetic algorithm model, the KNN model, and the XGBoost model, which are not discussed and applied in detail in this paper, due to limited space and time. New research methods and mathematical models are also being introduced, and future research can use better models for forecasting.

## 7. Conclusions

This paper adopts three models, namely logistic regression, neural network, and random forest, to construct a financial early warning model for Chinese listed shipping enterprises from both the micro- and macro-perspectives. The conclusions drawn through comparative analysis are as follows:

First, please note the significance of the micro-financial indicators. Among the selected 20 micro-financial indicators, the $X_6$ (current ratio), $X_7$ (quick ratio), $X_{15}$ (accounts payable turnover ratio), $X_{16}$ (total asset turnover ratio) four indicators show no significant difference between the crisis samples and the healthy enterprise samples. The four indicators did not pass the significance test; that is, they did not have sensitivity.

Second, the ranking of the index importance obtained by the random-forest model can discover the key factors affecting shipping companies. Among the micro early warning indicators, $X_1$, $X_4$, $X_9$, and $X_{20}$, r are the basic earnings per share, the operating income/operating profit, multiples of earned interest, and the net profit (the year-on-year growth rate), respectively; these have the greatest impact on whether the company is in financial crisis. Among the macro early warning indicators, $M_2$, $M_3$, CCFI, and the total amount of water transported by water in the country have the greatest impact on whether the company is in financial crisis [29].

Third, validity takes into account the economic cycle factors. The empirical results show that the model established by the samples from 2002 to 2009 has a good prediction effect on the samples from 2010 to 2015. Taking the macro-factors into account, the correct rates of the three models of logistic regression, the neural network, and random forest reached 79.2%, 74.03%, and 80.52%, respectively. That is, the division of the economic cycle in this paper and the consideration of economic cycle factors are shown to be effective in the process of establishing the model [30,31].

Fourth, we consider the accuracy of the model. From the data point of view, after adding macro-economic indicators, the forecast rate increased. This indicates that the more comprehensive the factors considered, the more accurate the model is. Although the correct prediction rates of the test samples decreased compared with the training samples, they were all above 70%, indicating that the models have certain prediction effects. The accuracy of the three models is compared, and the random-forest early warning model has the highest prediction accuracy, reaching 100% under both index systems; the test sample reaches 76.62% and 80.52% under the two index systems, respectively [32].

Based on the conclusions, the random-forest financial early warning model has shown strong advantages in terms of the degree of interpretation of the indicators, the accuracy of the model, and the operability.

For business managers, companies should establish a financial early warning model that is suitable for its own business. The financial early warning model is operable in enterprise risk management, which enables managers to anticipate potential financial risks that may occur in the future in advance in order to be fully prepared to deal with the risks.

For regulators, a comprehensive evaluation system for listed companies can be established. The regulators can implement dynamic monitoring of companies in the market through early warning models to prevent possible systemic financial risks in advance.

**Author Contributions:** Conceptualization, Y.L. and M.L.; methodology, M.L., R.X. and Y.L.; formal analysis, Y.L.; investigation, R.X.; resources, R.X.; writing—original draft preparation, Y.L.; writing—review and editing, M.L. and C.Y.; visualization, M.L.; supervision, C.Y. All authors have read and agreed to the published version of the manuscript.

**Funding:** This research was funded by Chongqing Social Science Planning Research Project (Grant Number 2022PY50); Henan Province Philosophy and Social Science Planning Research Project (Grant Number 2021CJJ123); Shandong Province Social Science Planning Research Project (Grant Number 19CQXJ29).

**Institutional Review Board Statement:** Not applicable.

**Informed Consent Statement:** Informed consent was obtained from all subjects involved in the study.

**Data Availability Statement:** The dataset analyzed during this study is available from the corresponding author upon reasonable request.

**Conflicts of Interest:** The authors declare no conflict of interest.

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
