# Peer review of "Sustainability of Shipping Logistics: A Warning Model"

_sustainability, doi:10.3390/su151411219_

Round 1

Reviewer 1 Report

Please see the attatchment.

Reviewer 2 Report

Deat Authors, 

Your paper is of interest to the journal Sustainability for its content and methodological approaches.

The paper is well structured and presented, and carefully referenced. There are improvements to be considered in the final sections in relation to paragraph n.6 "Limitations and Discussions". The discussion is a bit too brief and could be expanded by discussing the results in relation to the referenced literature. Limitations could be instead included in the conclusions and be used to define future research perspectives. It would be of interest to also expand the conclusions if any relevant policy and managerial implications are presented based on your analysis. 

Reviewer 3 Report

Review Report

On

SUSTAINABILITY OF SHIPPING LOGISTICS: A WARNING MODEL

Comments on Manuscript Number: 2315248

Here below is my review report summary on the title mentioned above.  

Recommendation: Major Revision

Title

The title is not clear and needs to be modified

Abstract

The study findings were not clearly presented in the abstract, and improvements are needed in terms of the methodology used. The type of research conducted is unclear, whether it is a research review, a case report, or another form of research. The authors also did not discuss the novelty of the study or provide recommendations that could be beneficial for other studies on sustainability in shipping logistics. Additionally, the objective of the study is not clearly stated.

Introduction

  The introduction lacks a clear and coherent chronological linkage between the concepts of Shipping logistics, and sustainability. To enhance the flow and understanding for readers, it is recommended to establish a logical progression that begins with a brief overview of Shipping logistics, and how it related to sustainability. This chronological connection will help readers grasp how these concepts have evolved over time and how they are interconnected.   

Rationale and Novelty: It is essential for the authors to include a clear rationale for conducting this study. They should explain the existing research gaps or problems that their research aims to address. Additionally, the authors should explicitly highlight the novelty of their study. This may include discussing how their research contributes to the existing literature by focusing on a specific context or population, employing unique methodologies, or providing novel insights into the relationship between digitalization and green innovation. By clearly stating the rationale and novelty, the authors can convey the importance and originality of their study to the readers.

Literature

The study has no literature section

Section 2 should be written as “Related Literature Review”

In literature, more empirical studies should be highlighted with their finding

Where is the Conceptual framework of the study? how could academic readers understand the dependent and independent variables of the study?

Methodology

The current information written in section 2 should be incorporated in section 3. At the same time, it needs to be specific and clear  

The research design, approach, and sample techniques should be clearly identified in order to provide a comprehensive understanding of the study. The current version of the study lacks chronological linking, which makes it challenging for readers to delve deeper into the content and follow the logical flow of information. It is important to establish a clear and coherent structure that facilitates the reader's engagement and comprehension of the research.

General comments

Overall, I understand that the lack of chronological linkage and the absence of logical connections between information have made it challenging to navigate through the paper. Furthermore, the absence of a manuscript formatting style has further contributed to the lack of clarity. I strongly advise the authors to carefully review and recheck both the format and content of the manuscript. Here are some suggested steps to improve the paper:

    Language and Clarity: Carefully review the language used throughout the paper to ensure clarity and coherence. Eliminate ambiguous or unclear statements and strive for a more concise and precise writing style.

Round 2

Reviewer 3 Report

once again, congratulations